# Hand Gesture Recognition Using Ultrasonic Array with Machine Learning

**DOI:** 10.3390/s24206763

**Published:** 2024-10-21

**Authors:** Jaewoo Joo, Jinhwan Koh, Hyungkeun Lee

**Affiliations:** 1Department of Electronic Engineering, Gyeongsang National University, Jinju 52828, Republic of Korea; wnwodn66@gnu.ac.kr; 2Department of Electronic Engineering, Engineering Research Institute, Gyeongsang National University, Jinju 52828, Republic of Korea; 3School of Computer and Information Engineering, Kwangwoon University, 20 Kwangwoon-ro, Nowon-gu, Seoul 01897, Republic of Korea; hklee@kw.ac.kr

**Keywords:** ultrasonic array, hand gesture recognition, artificial intelligence, convolutional neural network

## Abstract

In the field of gesture recognition technology, accurately detecting human gestures is crucial. In this research, ultrasonic transducers were utilized for gesture recognition. Due to the wide beamwidth of ultrasonic transducers, it is difficult to effectively distinguish between multiple objects within a single beam. However, they are effective at accurately identifying individual objects. To leverage this characteristic of the ultrasonic transducer as an advantage, this research involved constructing an ultrasonic array. This array was created by arranging eight transmitting transducers in a circular formation and placing a single receiving transducer at the center. Through this, a wide beam area was formed extensively, enabling the measurement of unrestricted movement of a single hand in the X, Y, and Z axes. Hand gesture data were collected at distances of 10 cm, 30 cm, 50 cm, 70 cm, and 90 cm from the array. The collected data were trained and tested using a customized Convolutional Neural Network (CNN) model, demonstrating high accuracy on raw data, which is most suitable for immediate interaction with computers. The proposed system achieved over 98% accuracy.

## 1. Introduction

In recent years, the importance of gesture recognition technology has rapidly increased, leading to diverse and active research in this field. For example, studies on exploring rich semantics for open-set action recognition [1], graph-based multimodal sequential embedding for sign language translation [2], mixed resolution networks with hierarchical motion modeling for efficient action recognition [3], audio-visual speech and gesture recognition by sensors of mobile devices [4], and gloss-driven conditional diffusion models for sign language production [5] demonstrate various application possibilities in the field of gesture recognition. These studies contribute to the advancement of technologies that recognize and interpret human gestures to interact with electronic devices; the importance of gesture recognition technology for controlling electronic devices without input devices such as touchscreens and keyboards is also increasing [6,7,8,9,10,11,12]. The main methods of gesture recognition technology include systems based on motion capture suits [13], camera-based systems [14], radio-wave-based systems [15], and systems utilizing ultrasonic sensors. To conduct research on high-accuracy hand gesture recognition systems, it is essential to use sensors suitable for hand gesture recognition. Therefore, it is important to perform a comparative analysis of the unique advantages and limitations of each system. Table 1 presents the characteristics of various gesture recognition systems.

Motion capture suits provide very high accuracy and demonstrate excellent performance, particularly in recognizing complex movements. However, they require users to wear the suit, and the high cost of equipment and complex installation processes limit their application in real-life situations. Camera-based gesture recognition systems enable freehand gesture recognition and boast high recognition rates in well-lit environments. However, these systems are sensitive to lighting conditions, leading to a sharp decline in recognition rates in darker settings. Additionally, there is a risk of privacy invasion for users. Radio-wave-based gesture recognition systems consume relatively low power, enable non-contact recognition, and can penetrate walls or obstacles, making them useful in certain environments. However, their systems have not yet reached commercialization, and the requirement for a radio license restricts their practical use.

Ultrasonic sensor-based gesture recognition systems have several significant advantages compared to these existing methods. First, ultrasonic sensors are cost-effective and can be easily integrated into various devices, particularly due to their low cost and compact size. Second, ultrasonic waves are not affected by lighting conditions, allowing for stable performance even in dark environments. Third, ultrasonic sensors utilize the piezoelectric effect to detect objects through non-contact and have excellent power efficiency for operation. These advantages of ultrasonic sensors can be applied across various fields. In smart home devices, ultrasonic sensors enable the control of appliances through hand gestures without touch or voice commands. Additionally, in the automotive industry, drivers can control various vehicle functions with simple hand gestures, enhancing safety while driving. Ultrasonic sensor-based gesture recognition technology provides a more suitable alternative for real-life applications compared to existing camera-based systems or motion capture suits, particularly because it does not require wearing a device and is unaffected by lighting conditions, allowing for use in diverse environments.

This research aims to freely measure hand gestures and obtain high-accuracy data. Therefore, an ultrasonic array was utilized. Ultrasound is defined as sound with a frequency above 20 kHz. Ultrasonic sensors do not generate noise issues because they use frequencies that exceed the human audible range. They are capable of non-contact detection of objects and measuring the distance and speed. Also, they are used in diverse fields, including engineering, medicine, automation systems, and gesture recognition [16]. In the field of gesture recognition, ultrasonic sensors are commonly used as a technology to detect simple movements in the form of transducers that generate ultrasonic waves using the piezoelectric effect [17,18,19,20,21,22]. However, the beam pattern of ultrasonic transducers is greater than 50° at −6 dB, and the main lobe has a wide beamwidth, resulting in poor angular resolution.

The wide beam width makes it challenging to recognize more than one object in a single beam. Additionally, the wide beamwidth results in the intensity of the beam attenuating with distance, which reduces accuracy when the object being measured is farther away. However, a wide beamwidth is advantageous for recognizing hand gestures as it must detect single hand movements. The issue of decreased accuracy due to weak beam intensity can be resolved by using an array of ultrasonic transducers.

There exist a variety of ultrasonic transducer arrays, including linear, square, and circular arrays [23,24]. In this research, ultrasonic transmitting transducers were arranged in a circular pattern to measure hand gestures freely in a wide area. Additionally, to uniformly receive signals measured at various distances, two amplifiers were connected to the receiving transducer. This setup aimed to measure hand gesture image data, facilitating accurate and immediate interaction with the computer. Afterward, to train the image data collected through the ultrasonic array, a CNN structure was designed using 2023b MATLAB’s Deep Network Designer. CNN is suitable for ultrasonic image analysis as it allows the design of various filters and detailed configurations [25]. The CNN model utilized in the experiment was analyzed through the construction of a two-stage serial, three-stage parallel, and one-stage serial structure.

The following outline illustrates the research that focuses on measuring seven hand gestures using an ultrasonic transducer array and accurately classifying them with a CNN model. This research is divided into four sections: Section 1 introduces the theoretical background of ultrasonic sensors. Section 2 explains the proposed method and its working principles. Section 3 presents the data measurement process and learning outcomes. Section 4 summarizes the research findings and conclusions.

## 2. Materials and Methods

### 2.1. Ultrasonic Circular Array

The beam area of an ultrasonic transducer array varies depending on its configuration. In this research, ultrasonic transducers were arranged in a circular array. Figure 1a illustrates the measurable beam area for hand gestures with a single ultrasonic transducer, while Figure 1b shows the measurable beam area for hand gestures with a circular ultrasonic transducer array.

In Figure 1a, *θ* represents the radiation angle of a single transducer, and *L* denotes the distance to the point where the array measures the hand. The diameter of the beam capable of measuring hand gestures is given by Equation (1).
(1)$$D=2\tan\left(\theta \right)L,$$

The ultrasonic array in Figure 1b consists of a total of 8 transmitting transducers arranged in a circular pattern with a 15 mm spacing between each. Additionally, the receiving transducer is positioned at the center of this circular ultrasonic array with a 10 mm spacing from the transmitting transducers. To reduce the impact of vibrations caused by the piezoelectric effect, the ultrasonic transducers were arranged at distances of 15 mm and 10 mm, based on sufficient spacing criteria to avoid vibration issues. When using a single transducer, as shown in Figure 1a, the emitted beam from a single transducer forms a 1D circular diameter. In contrast, the beam from the circular ultrasonic transducer array forms a 2D circular diameter. This array generates a beam pattern that can detect all hand gestures within the 2D circular area and measure all hand movements in the 3-dimensional space composed of the X, Y, and Z axes. Figure 2 illustrates the ultrasonic transducer array utilized in the experiment.

### 2.2. The Operating Principle of an Ultrasonic Array

In this research, the microcontroller (MCU) board used is the ESP32 D1 mini. The ESP32 model has a size of $3.1*3.9({cm}^{2})$. The ESP32 D1 mini model is small in size, causing minimal interference even when attached to the same PCB as the ultrasonic array. Additionally, it has numerous GPIO pins, making it suitable for operating the array.

Using Arduino IDE on the ESP32, the code was developed to control the ultrasonic array. Eight transmitting ultrasonic transducers were connected to the ESP GPIO pins and programmed to emit signals with a delay of 0.1 s. Additionally, all ultrasonic transducers were set to continuously generate and emit 40 kHz pulse waves. Ultrasonic signals transmitted at 0.1-s intervals are repeatedly transmitted into the air and reflected off the experimenter’s hand each time it moves. The reflectivity is determined by the difference in acoustic impedance between the air and the hand. The smaller the difference in acoustic impedance, the less ultrasound is reflected, and the larger the difference, the more ultrasound is reflected. Here, $Z_{1}$ represents the acoustic impedance of the hand, and $Z_{2}$ is the acoustic impedance of the air. The reflectivity *R* is given by the following formula.
(2)$$R=100{\left(\frac{Z_{1}-Z_{2}}{Z_{1}+Z_{2}}\right)}^{2},$$

Air has a very small acoustic impedance of 0.0004 (g/cm^2^-s × 10^−5^), while the human hand has a very large acoustic impedance of 1.70 (g/cm^2^-s × 10^−5^). Consequently, the reflectivity of ultrasonic waves between air and the human hand is over 99%.

Ultrasonic signals have a high reflectivity, causing most of the signals to be reflected. However, the strength of the signal that reflects off the hand and returns is attenuated with distance. Therefore, it is important to consider signal attenuation in order to accurately measure hand gestures. This attenuation of the signal varies depending on the measurement distance, operating frequency of the ultrasonic transducer, and properties of the medium. A is the signal strength at distance *d*, $A_{0}$ is the initial signal strength, α is the attenuation factor, and d is the measurement distance; when d is the measurement distance, the signal strength *A* at distance *d* is given by the following formula.
(3)$$A=A_{0}e^{-\alpha d},$$

$\alpha_{0}$ is the attenuation coefficient in air, and α, the attenuation coefficient at operating frequency f, is given by the following formula.
(4)$$\alpha ={\alpha_{0}f}^{n},$$

In the experiment, hand gestures were measured at distances of 10 cm, 30 cm, 50 cm, 70 cm, and 90 cm. To measure hand gestures accurately even at the 90 cm distance, the ultrasonic array was designed considering the attenuation coefficient α. Figure 3 is the circuit diagram of the ultrasonic array utilized in the research.

The signal reflected off the hand was received through the receiving transducer in Figure 3c. Subsequently, the received signal is amplified through a differential amplifier 3a. In this case, the amplification resistance can be adjusted using a variable resistor to account for the attenuation coefficient corresponding to the distance. Following this, it is further amplified using the non-inverting amplifier in Figure 3b. $V_{out1}\;$is the output voltage, $A_{d}$ is the differential amplifier gain, $V_{1}$ is the voltage at the inverting input terminal, and $V_{2}$ is the voltage at the non-inverting input terminal; the gain of the differential amplifier is given by the following Equation (5).
(5)$$V_{out1}=A_{d}\left(V_{1}-V_{2}\right),$$

The amplified $V_{out1}\;$from the differential amplifier is further amplified using a non-inverting amplifier. $R_{1}$ and $R_{2}$ are the resistances in the non-inverting amplifier. The final amplification value is given by the following Equation (6).
(6)$$V_{out2}=V_{out1}\left(1+\frac{R_{1}}{R_{2}}\right)$$

The signal, which has been amplified on two occasions following its reception, is stored in the ESP32.

### 2.3. Measurement of Hand Gesture Data for the Seven Classes and Deep Learning Training

The received signal is stored in ESP32 and transmitted to the computer via the serial communication port. The ultrasonic signal received through the serial communication is quantized into pixel values from 0 to 255 using JavaScript, and the measured ultrasonic waveform is visualized on the HTML screen.

The dataset presented in this research consists of the seven fundamental hand gestures with the highest applicability, as illustrated in Figure 4.

The hand gesture data used in the experiment were collected by one male and one female experimenter with different hand sizes and shapes. Measurements were taken at 10 cm, 30 cm, 50 cm, 70 cm, and 90 cm intervals from the ultrasonic array to measure 300 approaching hand gestures, 300 right-to-left hand gestures, 300 left-to-right hand gestures, 300 top-to-bottom hand gestures, 300 bottom-to-top hand gestures, 300 rotate diagonally up left hand gestures, and 300 rotate diagonally up right hand gestures at each measurement point. A total of 21,000 hand gesture data points were collected.

In this research, a total of 21,000 validation data sets were trained. Attempts were made to train using various existing network models, including AlexNet and GoogLeNet. However, due to their complex neural network models and lack of optimization for ultrasonic image data analysis, these models proved impractical as they required a long training time and achieved accuracies below 70%. To improve training speed and accuracy, a new neural network model was designed. The neural network used for deep learning is a CNN model with a two-stage serial, three-stage parallel, and one-stage serial structure, as shown in Figure 5.

The proposed CNN model utilizes filters of sizes 1 × 1, 3 × 3, and 5 × 5 in the convolutional layers, with a stride of 1 × 1. The max pooling layer has a pool size of 5 × 5 and a stride of 1 × 1. To reduce the training time for ultrasonic image analysis, optimized settings and a simple architecture have been employed. The CNN model used in the experiment is optimized for ultrasonic image analysis. To prove this, deep learning training was conducted using the same data with various network models. The data used for comparison and analysis consist of measurements taken at the midpoint of distances from 10 cm to 90 cm, at the 50 cm mark. The network models used for comparison include the proposed CNN model, AlexNet, GoogLeNet, ResNet-50, and VGG-19. Table 2 below shows the results of training 2100 images using various models under the same conditions.

The main specifications of the system that conducted learning are CPU: 11th Gen Intel (R) Core (TM) i7-11700 @ 2.50 GHz, RAM: 32.0 GB, and GPU: Geforce RTX3090.

## 3. Results

### 3.1. Experimental Measurement Process and Testing

In this research, the dataset used for experiments consists of seven classes as shown in Figure 4. The experiments utilized hand gesture data measured by two experimenters using an ultrasonic array at intervals of 20 cm from 10 cm to 90 cm. Figure 6 shows the process of measuring hand gestures.

The collected data underwent the process shown in Figure 7. A total of 21,000 raw data points were used for deep learning training. Additionally, 21,000 data points processed with 2D FFT and 21,000 data points that were normalized after 2D FFT processing were also used for deep learning training. All data were trained using the CNN model presented in Figure 5. Table 3 below shows the validation accuracy of hand gesture data measured by two experimenters at different distances.

As shown in Table 3, the two experimenters achieved the highest accuracy with raw data, which is essential for immediate interaction with computers. The raw data achieved over 97% accuracy at all measurement distances and demonstrated overwhelmingly higher accuracy compared to data processed through complex 2D FFT and normalization steps. The following Figure 8, Figure 9 and Figure 10 are confusion matrices showing test accuracy.

Figure 8, Figure 9 and Figure 10 show confusion matrices displaying test accuracies for seven hand gestures. The data used for testing were not utilized for validation. A total of 50 images per hand gesture were tested, with the test data measured at a distance of 50 cm by a male experimenter. Figure 9 and Figure 10, trained on preprocessed data, exhibit high accuracy for certain gestures but lower accuracy for others. In contrast, the confusion matrix of Figure 8, trained on raw data, demonstrates consistently high accuracy across all seven hand gesture classes.

### 3.2. Real-Time Testing

The raw data trained through deep learning were stored in the MATLAB workspace. The stored data were categorized by distance and hand gestures, comprising a total of 21,000 images. In the process of measuring hand gestures shown in Figure 6, when a new ultrasonic image was stored, it underwent real-time testing against a dataset of 21,000 pre-trained hand gesture data to classify the hand gesture class. Figure 11 illustrates real-time measurement of approaching hand gesture, identified as approach movements by comparing them with pre-trained data.

## 4. Conclusions

This research developed an ultrasonic array model for accurate hand gesture recognition. Hand gestures measured using the ultrasonic array were validated and tested using a CNN model optimized for ultrasonic image analysis. Seven fundamental hand gestures were subjected to deep learning, and this approach demonstrates significant advancements over existing solutions. Traditional gesture recognition systems, such as motion capture suits and camera-based methods, face limitations related to cost, environmental constraints, and user privacy. In contrast, the ultrasonic sensor-based system presented here offers a cost-effective and versatile alternative that operates effectively in diverse lighting conditions and does not require physical contact.

The arrangement of ultrasonic transducers in a circular pattern enhances the measurable spatial coverage and improves gesture recognition accuracy. This method overcomes the limitations of a single ultrasonic transducer, such as reduced measurement accuracy over distance and restricted detection range, by employing an array configuration. The real-time testing of new ultrasonic images against a comprehensive dataset of 21,000 pre-trained hand gesture data points showcases the system’s potential for immediate interaction with various applications, including smart home devices and automotive systems. Furthermore, as confirmed by two experimenters, the system achieved a high accuracy of over 97% across all measurement distances. The findings suggest that ultrasonic gesture recognition systems can serve as a reliable alternative in situations where traditional methods struggle due to environmental constraints or the involvement of various users.

There are opportunities to explore the practical applications of the ultrasonic array model in the future. For example, potential applications include contactless TV remote controls for hospital inpatient rooms that prioritize hygiene, and ultrasonic sensor canes to assist visually impaired individuals in their mobility. Overall, this research could lay the groundwork for the application of gesture recognition technology using ultrasonic sensors. It also has the potential to enhance human–computer interactions across various fields.

## Figures and Tables

**Figure 1 sensors-24-06763-f001:**
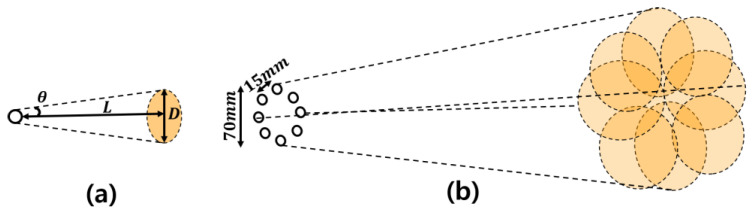
The measurable beam area for hand gestures with an ultrasonic transducer: (**a**) The measurable beam area for hand gestures with a single ultrasonic transducer; and (**b**) the measurable beam area for hand gestures with a circular ultrasonic transducer array.

**Figure 2 sensors-24-06763-f002:**
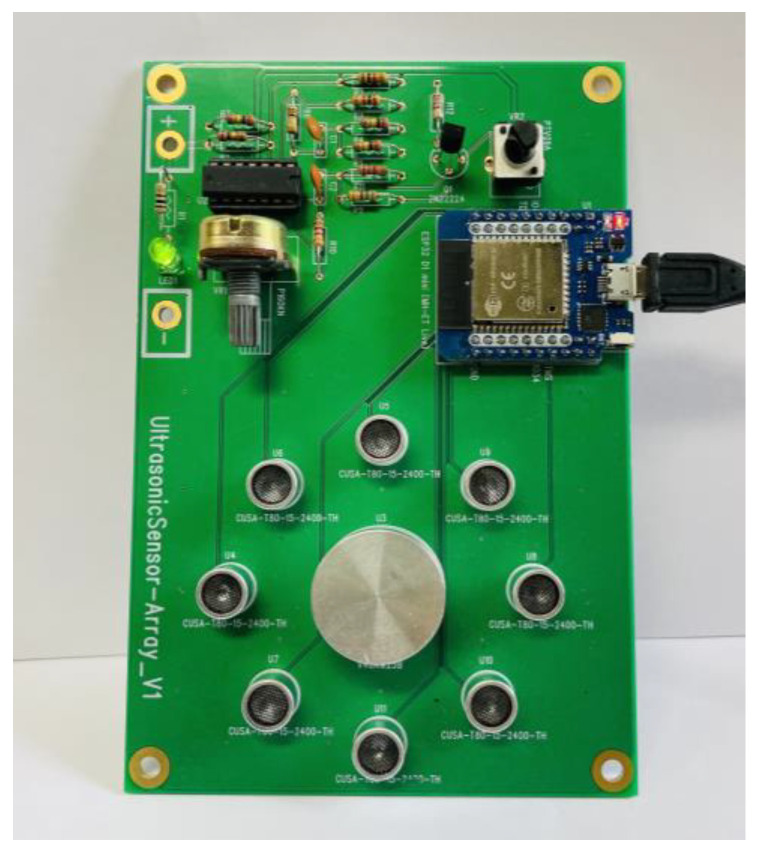
Ultrasonic transducer array utilized in the experiment.

**Figure 3 sensors-24-06763-f003:**
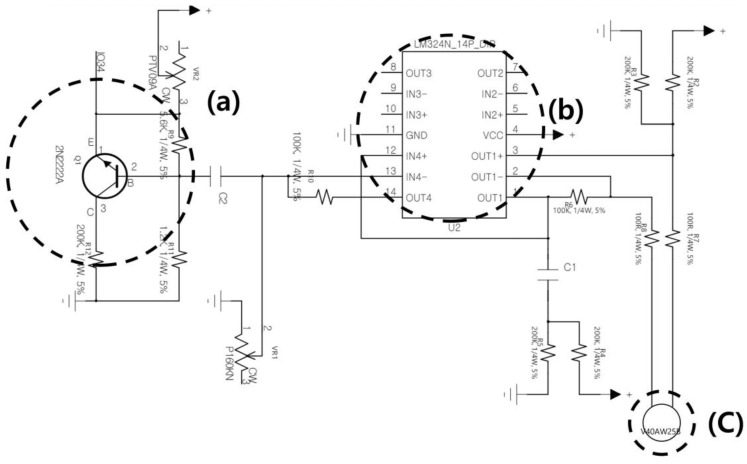
Ultrasonic array circuit diagram utilized in the research: (**a**) Differential amplifier that amplifies the received signal; (**b**) non-inverting amplifier that amplifies the received signal; and (**c**) receiving transducer.

**Figure 4 sensors-24-06763-f004:**
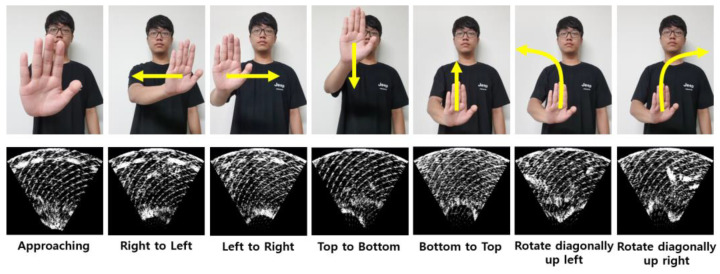
Seven hand gestures used in the experiment.

**Figure 5 sensors-24-06763-f005:**
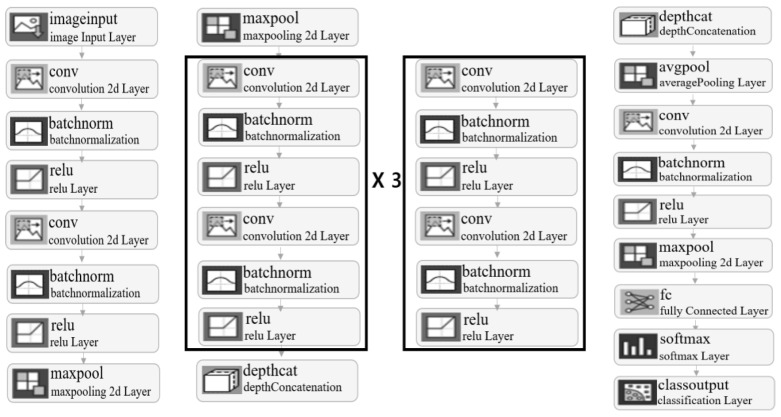
Block diagram of the proposed CNN model.

**Figure 6 sensors-24-06763-f006:**
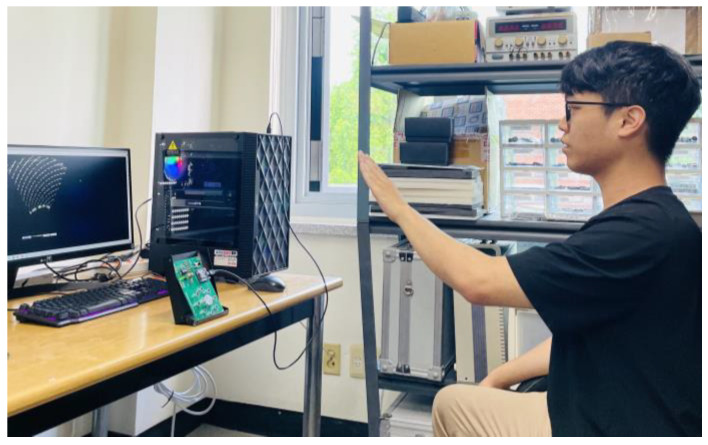
Process of measuring hand gestures.

**Figure 7 sensors-24-06763-f007:**
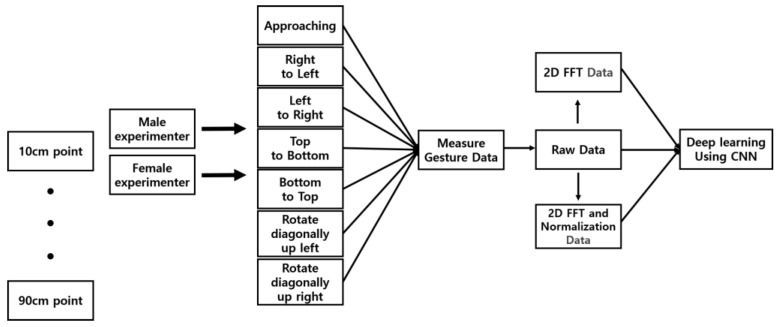
Data measurements and CNN training process.

**Figure 8 sensors-24-06763-f008:**
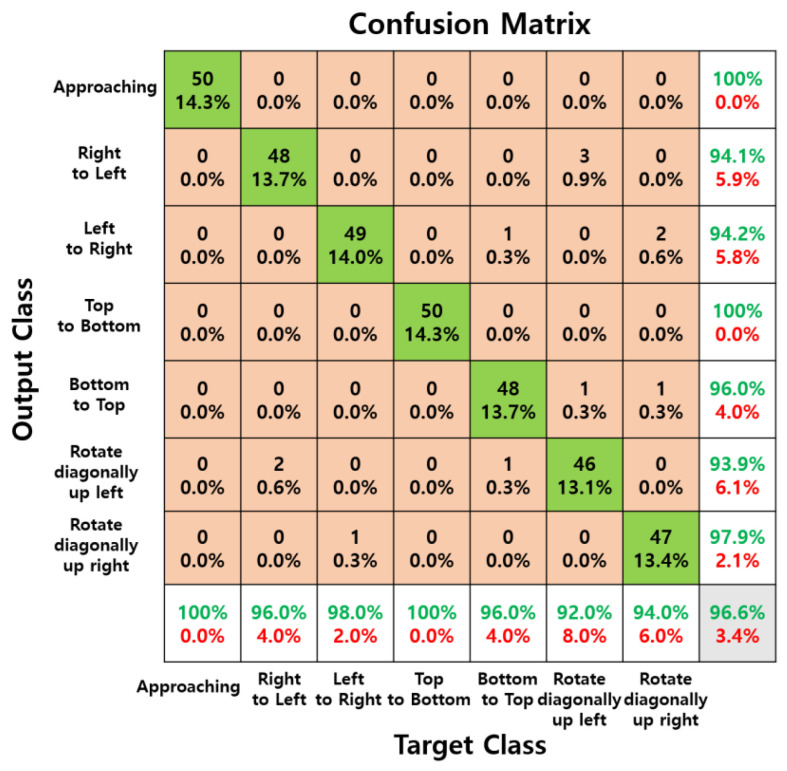
Confusion matrix of raw data measured for seven hand gestures at a distance of 50 cm, with 50 samples per class for testing.

**Figure 9 sensors-24-06763-f009:**
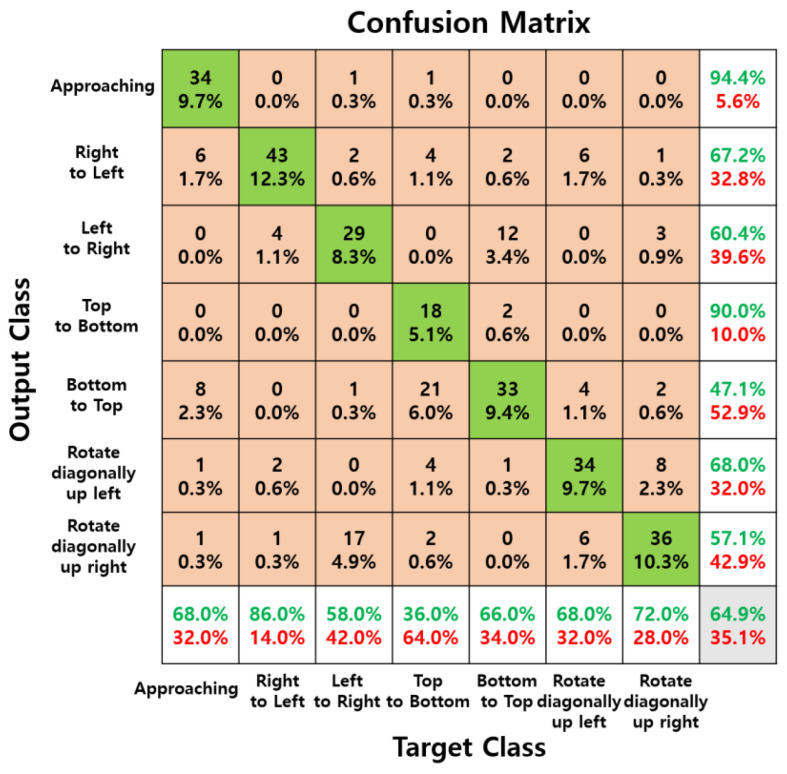
Confusion matrix of 2D FFT data measured for seven hand gestures at a distance of 50 cm, with 50 samples per class for testing.

**Figure 10 sensors-24-06763-f010:**
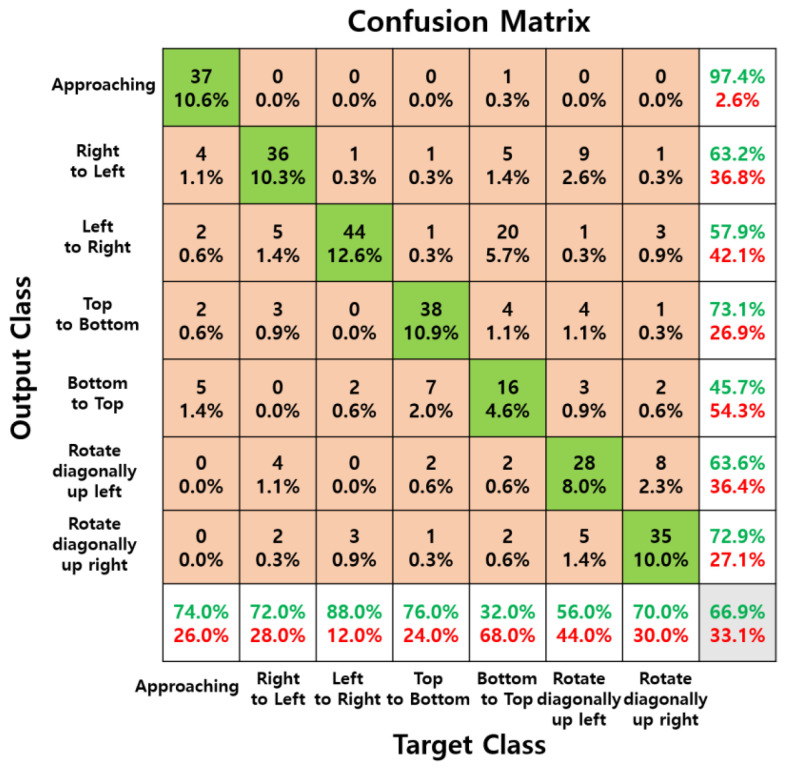
Confusion matrix of 2D FFT and normalization data measured for seven hand gestures at a distance of 50 cm, with 50 samples per class for testing.

**Figure 11 sensors-24-06763-f011:**
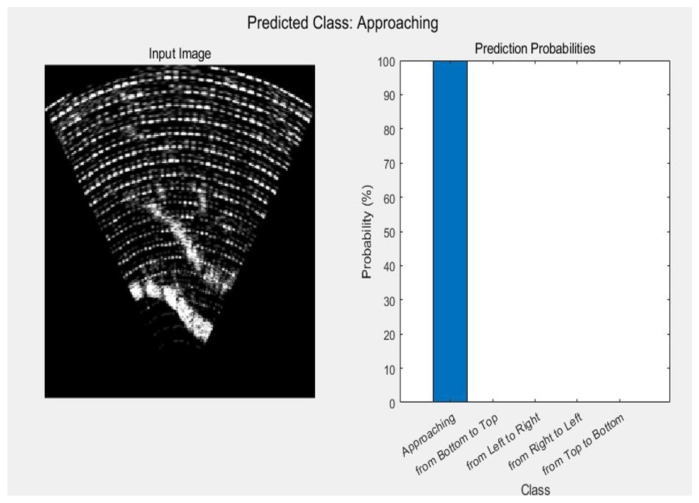
Real-time measurement of approaching hand gesture.

**Table 1 sensors-24-06763-t001:** The characteristics of various gesture recognition systems.

Sensor Type	Motion Capture Suits	Camera	Radio Wave	Ultrasonic
Accuracy	Very high	Medium	High	High
Price	Very expensive	Medium	Expensive	Cheap
Response Time	Very Fast	Medium	Fast	Fast
Environmental Constraints	High (suit wear required)	High (sensitive to lighting and background)	Medium (radio license required)	Low (not affected by lighting)

**Table 2 sensors-24-06763-t002:** Results of the comparison of validation accuracy and training time for the CNN model used in the experiment and four other models based on the same 2100 images.

Model	Accuracy (%)	Training Time (minutes)
Proposed CNN	97.14%	9 min
AlexNet	52.06%	24 min
GoogLeNet	56.03%	30 min
ResNet-50	69.05%	44 min
VGG-19	54.76%	46 min

**Table 3 sensors-24-06763-t003:** The validation accuracy of raw data, 2D-FFT data, and normalized data measured by two experimenters at 20 cm intervals.

Experimenter	Measure Distance (cm)	Validation Accuracy (%)		
		Raw Data	2D-FFT Data	Normalization Data
Male	10 cm	99.13%	68.35%	71.59%
	30 cm	98.87%	67.78%	67.67%
	50 cm	98.57%	67.11%	65.29%
	70 cm	97.44%	64.43%	66.16%
	90 cm	97.24%	62.75%	66.24%
Female	10 cm	99.08%	69.13%	70.44%
	30 cm	98.92%	66.22%	67.44%
	50 cm	99.02%	65.49%	64.52%
	70 cm	98.32%	63.32%	63.21%
	90 cm	97.13%	63.43%	63.29%

## Data Availability

The data will be made available to interested scientists upon request.

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
