# Peer review of "Hand Gesture Recognition Using Ultrasonic Array with Machine Learning"

_sensors, 2024, doi:10.3390/s24206763_

Round 1
Reviewer 1 Report
Comments and Suggestions for Authors
Thank you for the opportunity to review this article. The article is interesting and brings new perspectives on human-machine or human-computer interfaces that are in line with the Industry 5.0 concept. This method can be an interesting complement to existing gesture recognition solutions. However, a better comparison of the proposed solution with existing ones is lacking. The article needs some improvements.
Comments and suggestions:
1. The section on related works is very short. Absent, for example, is a discussion of gesture recognition systems that are based on motion-capture suits. Discuss the differences between camera-based gesture recognition systems, motion-capture suits, and your solution.
2. Discuss more broadly the benefits of your solution versus using camera systems or motion capture suits. What are the advantages of your solution? Where do you see its use? Why should they be used instead of conventional existing solutions?
3. The conclusion is very short. I recommend (e.g. in a bulleted list) to describe the main scientific or application benefits compared to existing solutions. Conclude with a summary of opportunities for further research.
After revision, the article can be judged again.
Author Response
- The section on related works is very short. Absent, for example, is a discussion of gesture recognition systems that are based on motion-capture suits. Discuss the differences between camera-based gesture recognition systems, motion-capture suits, and your solution.
We sincerely appreciate your valuable suggestion. In the revised manuscript, we have expanded the related works section to specifically discuss motion capture suits, camera-based systems, radio wave-based systems, and our ultrasonic sensor-based system. This additional content can be found on page 2, lines 46–58, where we have added a detailed comparison of gesture recognition systems in Table 1. This table compares various factors such as accuracy, cost, response speed, and environmental sensitivity. Through this expanded discussion, we have clarified the differences between currently used technologies and the ultrasonic-based gesture recognition system.
- Discuss more broadly the benefits of your solution versus using camera systems or motion capture suits. What are the advantages of your solution? Where do you see its use? Why should they be used instead of conventional existing solutions?
We have added an in-depth discussion on the advantages of our solution and potential applications on page 2, lines 59–72. The benefits of ultrasonic sensors, such as their low cost, high accuracy in low lighting conditions, and independence from wearable devices, have been emphasized. Additionally, we included application examples such as smart home control systems through simple hand gestures and automotive gesture recognition. Thanks to your valuable suggestion, we have significantly enhanced the quality of the paper by adding both the advantages of ultrasonic sensors and their potential applications. We sincerely appreciate it.
- The conclusion is very short. I recommend (e.g. in a bulleted list) to describe the main scientific or application benefits compared to existing solutions. Conclude with a summary of opportunities for further research.
Thank you for your suggestion to improve the conclusion. We have completely restructured the conclusion on page 10, lines 278–304. In this section, we scientifically compare and explain the methods used in existing gesture recognition systems with those using ultrasonic sensors. Additionally, we have included information about future research opportunities, such as applying ultrasonic sensors to non-contact TV remotes and developing ultrasonic sensor canes to assist visually impaired individuals.
Reviewer 2 Report
Comments and Suggestions for Authors
1. CNN structure is old, and although it is the first attempt at this task, the method itself is not novel. There are a lot of advanced and efficient network structures worth trying.
2. There is a lack of comparative experiments with different models. You can use the same data to train different models and compare their gesture recognition results to prove that the model you choose is the best
3. The paper mentions that gesture recognition methods include image processing, radio waves, and ultrasonic sensors. A comparison of the performance of the other two methods on the task of gesture recognition should be added to the article.
4. The data for training and testing is only five very simple gestures. There is no way to prove that the method in this paper can also perform well in complex gesture recognition tasks. Therefore, the variety of gestures should be expanded and the complexity of gestures should be increased.
5. The method of gesture recognition is updated quickly. The references cited are all from a long time ago. It is recommended to add some related works in recent years to this paper, such as [1*] [2*] [3*] [4*] [5*].
[1*] Exploring rich semantics for open-set action recognition. TMM 2023.
[2*] Graph-based multimodal sequential embedding for sign language translation. TMM 2022.
[3*] Mixed Resolution Network with hierarchical motion modeling for efficient action recognition. KBS 2024.
[4*] Audio-visual speech and gesture recognition by sensors of mobile devices. Sensors 2023.
[5*] Gloss-driven Conditional Diffusion Models for Sign Language Production. TOMM 2024.
Comments on the Quality of English LanguageModerate editing of English language required.
Author Response
- CNN structure is old, and although it is the first attempt at this task, the method itself is not novel. There are a lot of advanced and efficient network structures worth trying.
Thank you for your important observation. Following your suggestion, we conducted additional experiments using various network architectures, including GoogLeNet, ResNet-50, VGG-19, and AlexNet. This information has been added on pages 6 and 7, lines 213–224.
- There is a lack of comparative experiments with different models. You can use the same data to train different models and compare their gesture recognition results to prove that the model you choose is the best
We sincerely appreciate your valuable suggestion. We conducted additional comparative experiments on several network models using the same dataset of 2,100 ultrasonic images, and the results are detailed in Table 2 on page 7. Our research aims for accurate gesture recognition, so we selected high accuracy as a key comparison criterion. Additionally, since we need to train on 21,000 image data points, we also chose training speed as a primary comparison criterion. Despite its simplicity, our model outperformed other models in both accuracy and training time, further justifying its selection.
- The paper mentions that gesture recognition methods include image processing, radio waves, and ultrasonic sensors. A comparison of the performance of the other two methods on the task of gesture recognition should be added to the article.
We sincerely appreciate your valuable suggestion. We have added a thorough comparison of the performance of gesture recognition systems using motion capture suits, image processing systems through cameras, radio wave-based systems, and ultrasonic sensors. This comparison can be found in Table 1 on page 2. The table presents a comparison based on accuracy, cost, response speed, and environmental sensitivity. Additionally, we have included detailed information on the performance differences between traditional gesture recognition methods and those utilizing ultrasonic sensors in lines 48–72 on page 2.
- The data for training and testing is only five very simple gestures. There is no way to prove that the method in this paper can also perform well in complex gesture recognition tasks. Therefore, the variety of gestures should be expanded and the complexity of gestures should be increased.
Thank you for pointing this out. We have expanded the dataset to include seven gestures by adding two complex movements: Rotate diagonally up left and Rotate diagonally up right, thereby increasing the complexity of the recognition task. The new gestures are described in Figure 4 on page 6, and in lines 197–199. The corresponding performance results are reflected in Table 3 on page 8. Additionally, the test accuracy for each of the seven gestures is shown in Figures 8, 9, and 10 on pages 8 and 9. This expansion demonstrates the robustness of our system in handling more complex gestures.
- The method of gesture recognition is updated quickly. The references cited are all from a long time ago. It is recommended to add some related works in recent years to this paper, such as [1*] [2*] [3*] [4*] [5*].
[1*] Exploring rich semantics for open-set action recognition. TMM 2023.
[2*] Graph-based multimodal sequential embedding for sign language translation. TMM 2022.
[3*] Mixed Resolution Network with hierarchical motion modeling for efficient action recognition. KBS 2024.
[4*] Audio-visual speech and gesture recognition by sensors of mobile devices. Sensors 2023.
[5*] Gloss-driven Conditional Diffusion Models for Sign Language Production. TOMM 2024.
Thank you for your suggestion to include more recent research. We have updated the references section and added the recent papers you recommended. These citations have been added on page 1, lines 29–38.
Round 2
Reviewer 1 Report
Comments and Suggestions for Authors
The article can be accepted now.